

# Remote sensing captures varying temporal patterns of vegetation between human-altered and natural landscapes

Misha Leong[1] and George K. Roderick[2]

[1] Department of Entomology, California Academy of Sciences, San Francisco, CA, United States of America
[2] Department of Environmental Science, Policy and Management, University of California, Berkeley, CA, United States of America

## ABSTRACT

Global change has led to shifts in phenology, potentially disrupting species interactions such as plant–pollinator relationships. Advances in remote sensing techniques allow one to detect vegetation phenological diversity between different land use types, but it is not clear how this translates to other communities in the ecosystem. Here, we investigated the phenological diversity of the vegetation across a human-altered landscape including urban, agricultural, and natural land use types. We found that the patterns of change in the vegetation indices (EVI and NDVI) of human-altered landscapes are out of synchronization with the phenology in neighboring natural California grassland habitat. Comparing these findings to a spatio-temporal pollinator distribution dataset, EVI and NDVI were significant predictors of total bee abundance, a relationship that improved with time lags. This evidence supports the importance of differences in temporal dynamics between land use types. These findings also highlight the potential to utilize remote sensing data to make predictions for components of biodiversity that have tight vegetation associations, such as pollinators.

## INTRODUCTION

Increasing temperatures as a result of global climate change have led to shifts in phenology for many species (*Hughes, 2000*; *Walther et al., 2002*; *Menzel et al., 2006*; *Parmesan, 2006*), and widespread debate over the consequences of critical interaction mismatches (*Tylianakis et al., 2008*; *Hegland et al., 2009*; *Bartomeus et al., 2011*; *Willmer, 2012*). However, not all phenological change is the direct result of changing climate. Land use change, such as urbanization and agricultural expansion, includes the deliberate introduction of novel plants into communities. These plants, both exotic ornamentals and crops, are often accompanied by watering and supplemental nutrient inputs that extend survival potential in the targeted landscape, leading to different flowering seasons. Therefore, on a plant community-scale, different land use types can be expected to experience distinct patterns of phenological change.

Corresponding author
Misha Leong,
mleong@calacademy.org

Land surface phenology is the timing of overall surface vegetation growth (*Morisette et al., 2009*), and differs from traditional definitions of vegetation phenology (i.e., species specific life cycle events). Major advancements in assessing phenology on a landscape scale have been made possible through the use of satellite products and improved cyber-infrastructures (*Reed et al., 1994*; *Zhang et al., 2003*; *De Beurs & Henebry, 2005*; *Morisette et al., 2009*). Remote sensing techniques have provided tools to detect land-surface phenology, such as data from NASA's moderate-resolution imaging spectroradiometer (MODIS), which are being used to produce 250 m spatial resolution products every 16 days (https://lpdaac.usgs.gov/data_access).

Differences in land surface phenology have been detected as a result of land use change (*De Beurs & Henebry, 2005*; *Buyantuyev & Wu, 2012*; *Kariyeva & Van Leeuwen, 2012*; *Neeti et al., 2012*). In some cases, the changes in land surface phenologies between years have given insight into the history of regions that have experienced socio-economic and geopolitical transitions, such as changes in irrigation regimes in central Asia after the fall of the Soviet Union (*Kariyeva & Van Leeuwen, 2012*), or the expansion of urbanization in parts of Mexico (*Neeti et al., 2012*). Differences in phenology between neighboring land use types within a year also provide insights, such as vegetation phenology of urban landscapes found to be out of synchrony with patterns of phenology in the surrounding desert in the American southwest (*Buyantuyev & Wu, 2012*).

As a result of land use change, such phenological differences in vegetation could lead to phenological differences in other groups of organisms, especially if plants exert bottom-up control on the organisms that interact with them. Because primary productivity can be linked to biodiversity (*Waide et al., 1999*; *Dodson, Arnott & Cottingham, 2000*; *Chase & Leibold, 2002*), there exists great potential to use remote sensing of primary productivity data as a way to predict biodiversity (*Kerr & Ostrovsky, 2003*; *Duro et al., 2007*). However, early efforts to apply this technique were less powerful than expected (*Nagendra, 2001*). Some evidence suggests that remote sensing data can predict biodiversity; for example, peak vegetation indices in multiple studies are correlated with higher avian diversity (*Jorgensen & Nohr, 1996*; *Bino et al., 2012*), although in other systems the relationship is less clear (*Duro et al., 2007*).

Linking remote sensing data with biodiversity has been limited despite its great potential (*Nagendra, 2001*). One limitation may be that, in many comparisons, vegetation indices are treated statically, rather than as temporally dynamic (*Bino et al., 2012*). Indeed, use of a multi-season within year vegetation index was found to be a much more accurate predictor of biodiversity (*Nagendra, 2001*; *Krishnaswamy et al., 2009*). Another limitation is the scale at which remote sensing data are able to uncover patterns—these indices may be more suitable across landscape types with more dramatic differences in vegetation, such as human-altered landscapes.

Finally, measuring biodiversity may be too complex, and instead, remote sensing data may be more likely to resolve patterns for those organisms that exhibit tight linkages with plant communities for which tight linkages with vegetation indices are known.

Bees are a key pollinator group with close vegetation associations, since bees strongly depend on flowers for both nectar and pollen. Bees provide the majority of animal-mediated pollination services on which an estimated 87.5% of flowering plants depend (*Ollerton, Winfree & Tarrant, 2011*). The value of pollination in agriculture is estimated at $200 billion worldwide (*Gallai et al., 2009*), due to many foods that are essential for food security and a healthy human diet, including numerous fruits, vegetables, and nuts.

Bees are closely linked to floral availability in their environment. However, the temporal dynamics of floral resources can vary between land use types. In California grasslands, for example, there is typically a large burst of blooming in the spring, which tapers off in the summer. Neighboring urban areas often have ornamentals enhanced with external inputs, resulting in a steady patterns with only minor changes throughout the year, while agricultural landscapes have booms and busts of flowering that follow the pattern of local crops.

Here, we analyze how vegetation phenology varies in a human-altered California grassland landscape. Specifically, we ask whether human-altered landscapes in California grasslands experience phenological diversity that is out of synchrony with that of surrounding natural areas. We then explore how these changes in phenology correlate with those of the bee community that depends on floral resources.

## METHODS

### Study region

The study system was located around Brentwood, in east Contra Costa County, California, where agricultural, urban, and natural areas intersect with each other within a 50 × 50 km region. East Contra Costa County has had a farming community presence since the late 19th century. The agricultural areas of Brentwood, Knightsen, and Byron mostly consist of orchards (cherries, stone fruit, grapes and walnuts), corn, alfalfa, and tomatoes (*Guise, 2011*).

Due to a housing boom in the 1990s, there was rapid residential growth in the area, causing the city of Brentwood to expand from less than 2,500 people in the 1970s to over 50,000 today (http://www.census.gov/2010census/). Nearby, Antioch now has over 100,000 residents (2010 US Census). A county water district (Los Vaqueros Watershed), regional park district (East Bay Regional Parks: Black Diamond Mines, Round Valley, and Contra Loma), and California state park (Mount Diablo) also all fall within the region, leaving large areas of land protected from development. This protected (hereafter referred to as "natural") land consists mainly of grasslands and oak woodlands, some portions of which are managed for grazing.

### EVI and NDVI data processing

Remote sensing data included MODIS land subset product MOD13Q1, NDVI (Normalized Difference Vegetation Index) and EVI (Enhanced Vegetation Index) data. MOD13Q1 is produced on a 16 day frequency and 250 m spatial resolution. NDVI and EVI are the two most frequently used vegetation index products, and are calculated using different algorithms based on detected light bands. We obtained EVI and NDVI geotif files through the

*Oak Ridge National Laboratory Distributed Active Archive Center (2011)*, using a query from 2000 to 2014 in a 50 × 50 km zone encapsulating the study region. We calculated and plotted NDVI and EVI time series averages and standard deviations for each of the three land use types using all available data files from 2000 to 2014 ($n = 320$). We tested for differences in the mean standard deviations and ranges between land use types by fitting ANOVA models followed by Tukey Honest Significant Difference tests at the 95% confidence level.

## Bee community dataset

We used the bee dataset from *Leong (2014)*, and describe the collection of the data in detail here. This dataset includes 21,874 specimens from 91 bee species groups that were sampled at multiple time points from 2010 to 2012 from urban, agricultural, and natural sites. In 2010, there were 18 sites, with 6 each classified as types "Urban", "Agricultural", and "Natural." In 2011 and 2012, sites were increased to a total of 24 sites, with 8 of each land use classification. Land use type classification was determined using NOAA's 2006 Pacific Coast Land Cover dataset (developed using 30 m spatial resolution Landsat Thematic Mapper and Landsat Enhanced Thematic Mapper satellite imagery). A 500 m buffer was created around each site, and the number of pixels classified as agricultural, urban, natural, water, or bare land was extracted. Some categories were grouped within NOAA's classification scheme with the following definitions for urban, agricultural, and natural: Urban- "High Intensity Developed", "Medium Intensity Developed", "Low Intensity Developed", and "Developed Open Space"; Agricultural- "Cultivated", "Pasture/Hay"; Natural- "Grassland", Deciduous Forest", "Mixed Forest", "Scrub/Shrub". Sites were classified categorically based on the dominant land use type (>50%).

Sites were selected to be at least 1 km away from all others, based on maximum assumed bee foraging ranges (*Gathmann & Tscharntke, 2002*). Although certain bee species have been recorded foraging as far as 1,400 m (*Zurbuchen et al., 2010*), most bees have nesting and foraging habitat within a few hundred meters of each other (*Gathmann & Tscharntke, 2002*; *Zurbuchen et al., 2010*). Sites were selected to be in easily accessible, open areas that received full sun. Natural areas were in grassland habitat, agricultural sites were either weedy field margin edges or fallow fields, and urban sites that were vacant lots or greenways. The human-altered sites were deliberately selected to not be adjacent to any mass-flowering crops or gardens.

At each site bee samples were obtained using a standardized pan trapping transect, each consisting fifteen 12 ounce bowls filled to the brim with soapy water (0.5 tablespoon of Blue Dawn dishwasher detergent diluted in 1 gallon of water) and spaced 5 m apart in alternating colors of fluorescent yellow, blue, and white (*LeBuhn et al., 2003*). Transects were set up and run for a 4 h period between 10:30am to 2:30pm (+30 min) in 2010, with 4 sites sampled per day, and all sites sampled on consecutive days. These 2010 transects were run twice, once in the early summer, and once in the late summer. However, sampling was modified in 2011 and 2012 for transects to be set up for a 24 h period, so that more sites could be run simultaneously and collections made more often. All 24 sites were sampled

within two consecutive collecting windows of 24 h, and were run in early spring, late spring, early summer, and late summer, for a total of 4 times per year.

Bees were collected from the pan traps by using a metal strainer, rinsed with water, frozen overnight or longer, and then pinned and labeled. Specimens were sorted to the genus level, and then to the species level with the assistance of Dr. Robbin Thorp (Professor emeritus, University of California, Davis). The only exception to identification at the species level were bees of the genus *Lasioglossum*, due to their overwhelming abundance, limited availability of taxonomic expertise for this group, and lack of known ecological diversity. Voucher specimens and the majority of the total collection were deposited at the Essig Museum of Entomology at University of California, Berkeley (http://essig.berkeley.edu).

### Analyses between EVI and NDVI variables and bee community

There were a total of 228 collecting events (2010: 6 sites of each land use type sampled twice; 2011–2012, 8 sites of each land use type sampled 4 times each). For each collecting event, we calculated the total bee abundance and number of species groups (species richness). For each bee collecting event, the closest MOD13Q1 date of collection was identified. NDVI and EVI values were extracted via bilinear interpolation at the collecting point for the closest MOD13Q1 date of collection, as well as the immediately previous three MOD13Q1 dates of collection (T-16, T-32, and T-48) to incorporate the potential effect of time lags. Some studies have demonstrated the significant impact of time lags between changes to a landscape and the subsequent ecological responses (*Caperon, 1969*; *Metzger et al., 2009*). Eight possible vegetation indices were represented by NDVI and EVI obtained at the closest date of collection to bee sampling event, as well NDVI and EVI at the previous 3 dates of collection.

All analyses were done in R 3.1.2 (*R Development Core Team, 2014*). We tested for the relative significance of NDVI and EVI values on total bee abundance and species richness with generalized linear mixed models. Vegetation index, land use type, their interaction, land use collecting method (4 h vs. 24 h pan traps), and day of year were designated as fixed effects, and site and year as random effects. Because land use type is categorical, we designated natural land use type as the baseline against which urban and agricultural land use types were compared. All analyses were fit with Poisson distributions.

## RESULTS

### Phenological diversity of vegetation across land use types

Urban, agricultural, and natural land use types underwent different phenological patterns of vegetation "green-up", with regards to their EVI and NDVI (Fig. 1). In this region, EVI and NDVI are roughly equivalent (although EVI values are consistently slightly lower), so for the purposes of simplicity, only EVI analyses are presented. From 2000 to 2014 at each 16 day interval, the average EVI for all pixels classified as each land use type underwent distinctly different, and consistent patterns (Fig. 2). Natural areas had a large burst of high values in the early spring and then trailed off, urban areas were relatively constant, and

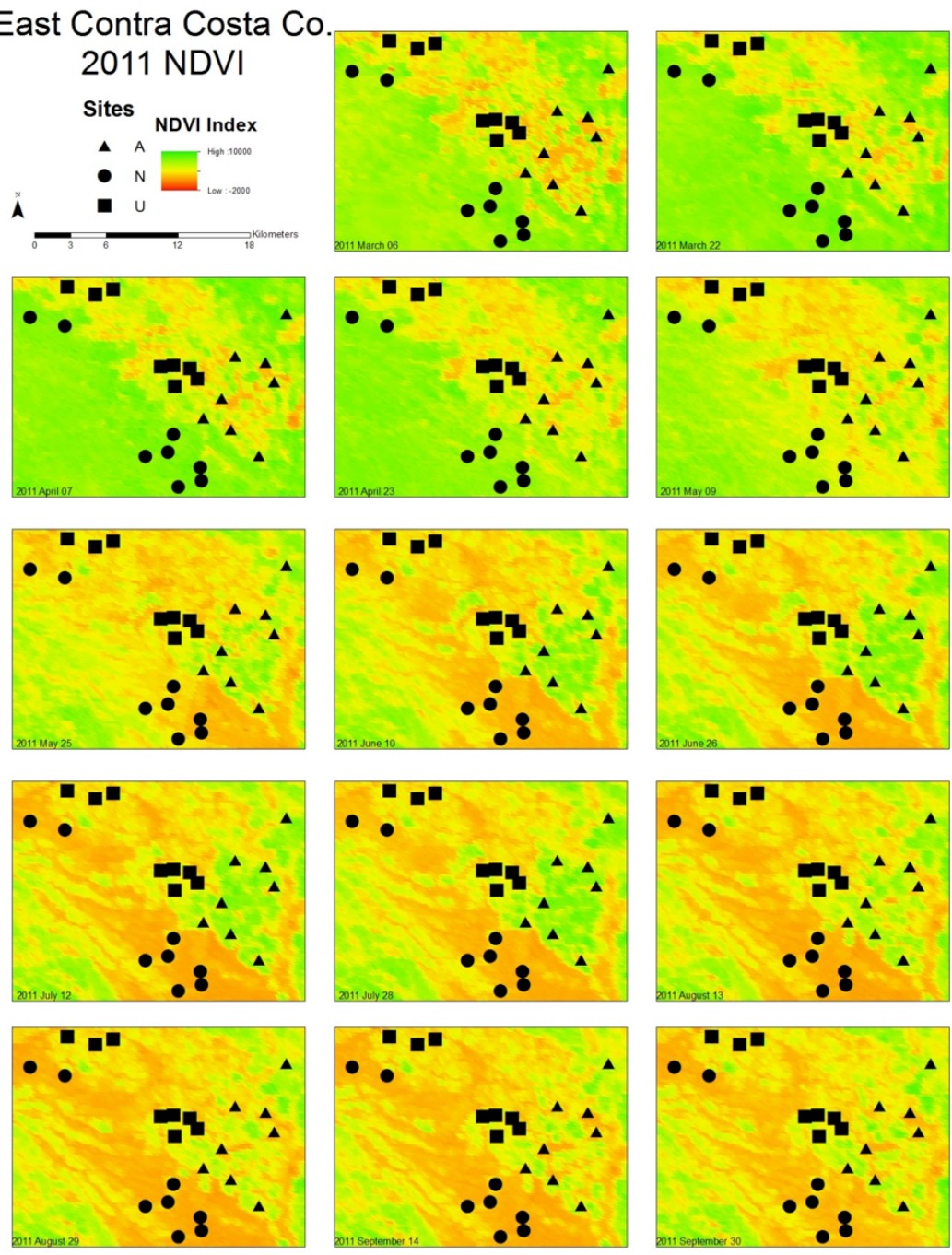

**Figure 1** **Maps illustrating the change in NDVI from March to September in East Contra Costa County, California.** A subset of bee collection sites (eight of each land use type) are marked on the map to illustrate the distribution of land use types. Triangles represent agricultural sites, squares represent urban sites, and circles represent natural sites. As the year progresses, higher NDVI values are associated with different land use types.

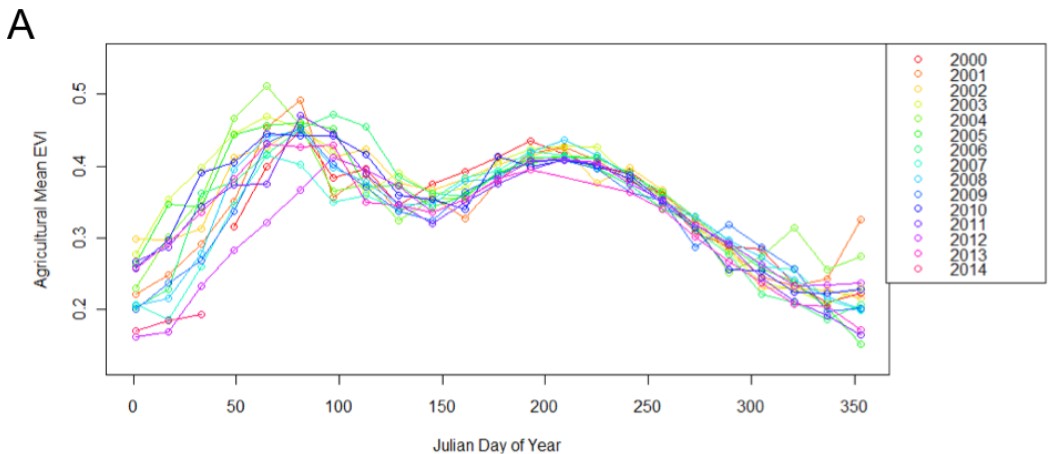

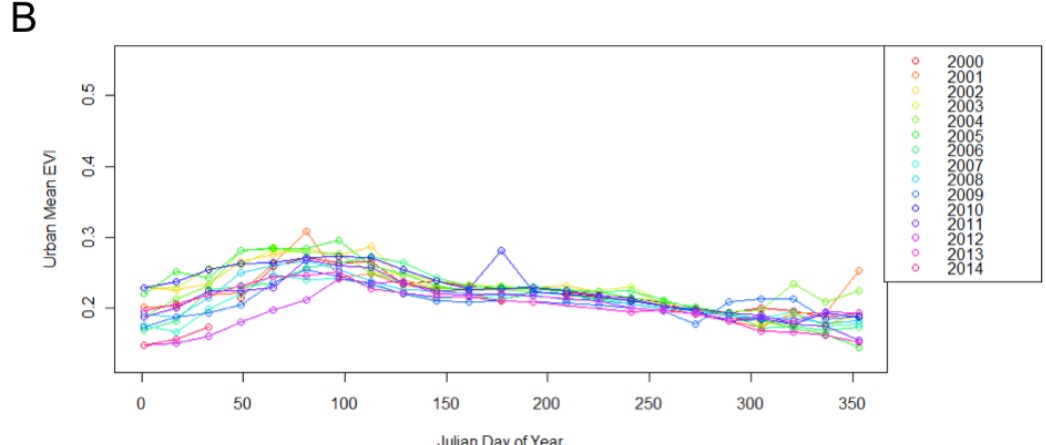

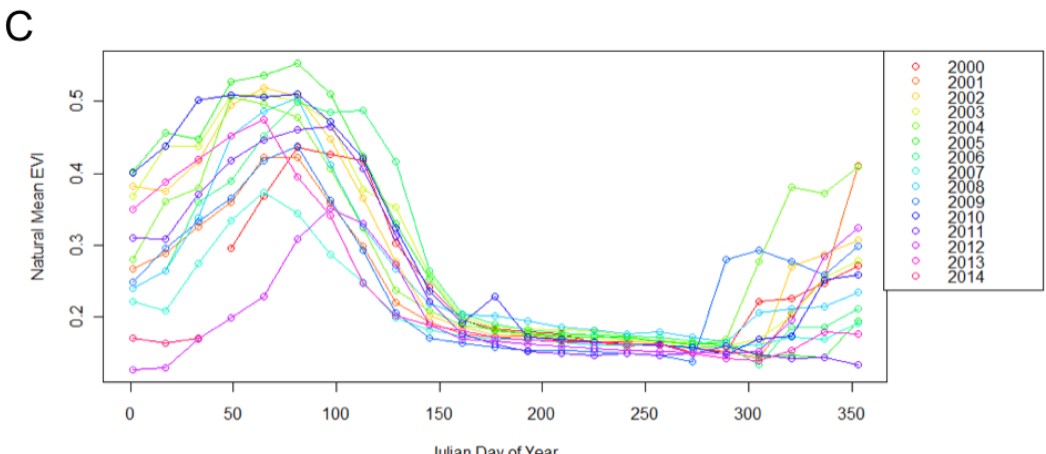

**Figure 2  Different land use types exhibit different patterns of change throughout the year, from 2000 to 2014.** Each point represents the mean EVI for all pixels of the same land use classification within the 50 × 50 km region encapsulating the study site. Agricultural sites (A), have two peak EVI values, urban sites (B) remain relatively even, and natural sites (C) have one peak EVI value.

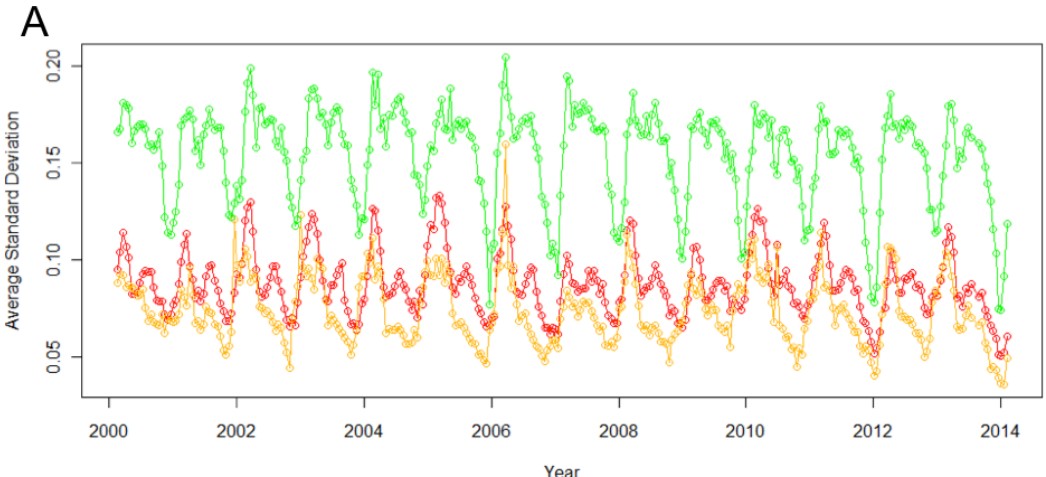

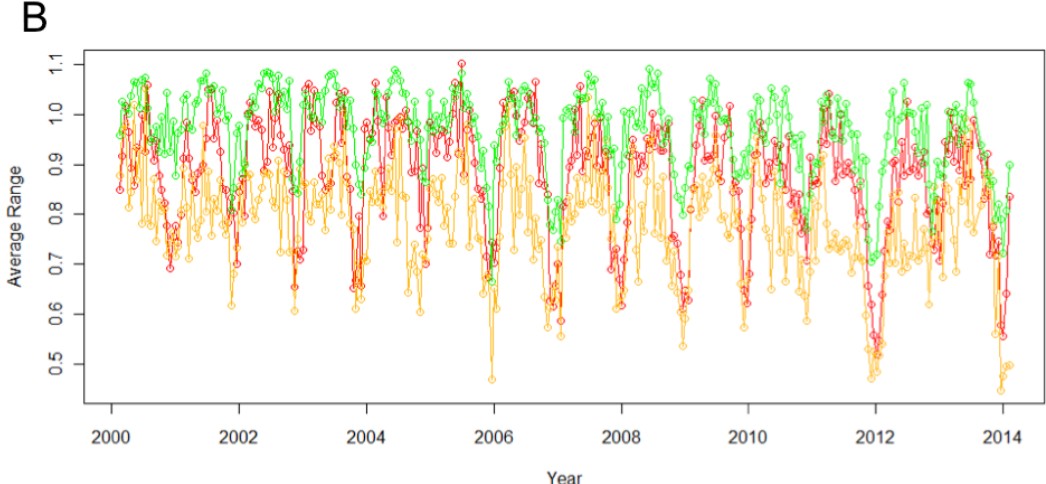

**Figure 3 Standard deviation (A) and range (B) of EVI across different land use types vary significantly.** Agricultural sites (green) consistently have the highest standard deviations and ranges, natural (yellow) have the lowest, and urban is in between (red).

agricultural areas had two greening peaks, the first at the same time as natural, and the second in the middle of the summer. Standard deviations (Tukey HSD, $p < 0.001$) and range (Tukey HSD, $p < 0.001$) were both larger in the human-altered landscapes than in natural areas (Fig. 3).

## Vegetation and bee community indices

For most models of total bee abundance, there was a significant impact of land use type, vegetation index value, and an interaction between the two (Tables 1 and 2). AIC values decreased, indicating better model fit, as the time lag of vegetation data increased to 48 days prior to the collecting event. Similar patterns were not found for species richness, and there was little to no relationship between vegetation indices and species richness.

When examining the data graphically, the relationship between higher vegetation indices and higher bee abundance was generally positive. However, a time lag of 16 days

Leong and Roderick (2015), *PeerJ*, DOI 10.7717/peerj.1141

**Table 1  Output of GLMM models for effect of EVI with varying time lags for total bee abundance.** Displayed are effect sizes, errors, and *p*-values for land use types (agricultural and urban) and EVI value. The last two columns denote whether or not there was a significant interaction (and in which direction) between each land use type and day. All significant explanatory variables with *p*-values <0.05 are in bold.

| Total bee abundance | AGR effect | AGR error | AGR *p*-value | URBAN effect | URBAN error | URBAN *p*-value | EVI effect | EVI error | EVI *p*-value | AGR × EVI | URB × EVI |
|---|---|---|---|---|---|---|---|---|---|---|---|
| Closest collecting period | 0.03703 | 0.21186 | 0.86126 | 0.18141 | 0.24058 | 0.45081 | **0.10862** | **0.01371** | **<0.001** | – | – |
| Closest collecting period—16 days | **1.09037** | **0.21596** | **<0.001** | **0.6076** | **0.24396** | **0.0128** | 0.40918 | 0.01211 | **<0.001** | – | – |
| Closest collecting period—32 days | **0.61977** | **0.20883** | **0.003** | **0.56755** | **0.23373** | **0.0152** | 0.44136 | 0.00947 | **<0.001** | – | – |
| Closest collecting period—48 days | 0.38849 | 0.20316 | 0.0558 | 0.105908 | 0.22515 | 0.6381 | **0.368123** | **0.00807** | **<0.001** | – | – |

**Table 2 Output of interaction effects between land use type and EVI value for GLMM models with varying time lags for total bee abundance.** Displayed are effect sizes, errors, and *p*-values for land use types (agricultural and urban) and EVI value interaction effect. All significant explanatory variables with *p*-values <0.05 are in bold.

| Total bee abundance | AGR × EVI effect | AGR × EVI error | AGR × EVI *p*-value | URB × EVI effect | URB × EVI error | URB × EVI *p*-value |
|---|---|---|---|---|---|---|
| Closest collecting period | −0.21171 | 0.01755 | <0.001 | −0.31618 | 0.03469 | <0.001 |
| Closest collecting period—16 days | −0.54888 | 0.01782 | **<0.001** | −0.42078 | 0.03409 | **<0.001** |
| Closest collecting period—32 days | −0.3695 | 0.01553 | <0.001 | −0.34271 | 0.02947 | <0.001 |
| Closest collecting period—48 days | −0.275724 | 0.014436 | <0.001 | −0.168625 | 0.023629 | <0.001 |

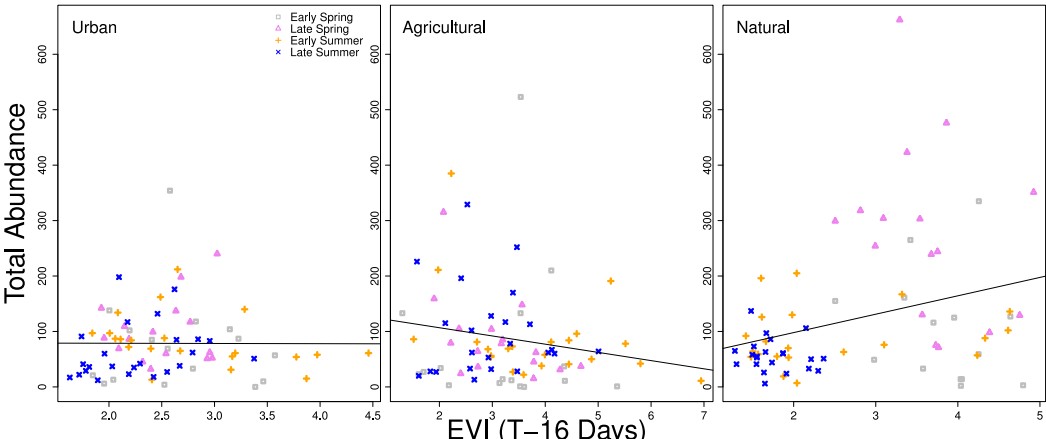

**Figure 4 Relationship between total bee abundance and EVI across seasons.** In this example, total bee abundance is correlated with the EVI value from the same location 16 days before the closest day of collection. Each panel represents a different land use type. Different seasons (early spring, late spring, early summer, and later summer) are plotted with different symbols. A simple linear regression of all points within each panel is fit, but is only statistically significant for the natural land use type ($p < 0.001$).

between the remote sensing data and the closest sampling date of bee collection exhibited differences in patterns between land use types. For natural sites, there was a significant positive correlation slope between EVI and higher bee abundance (when fit with a simple linear regression model, $p < 0.001$). The slope for bee abundance in urban sites was close to zero, and was a negative slope in agricultural sites, although neither were statistically significant (Fig. 4). Additionally, urban sites had a high amount of overlap between seasons in the correlations between vegetation indices and total abundance. Agricultural sites also had overlap between seasons, but overall variation was higher than with urban sites. Natural sites had little overlap between months and correlations were more dispersed, particularly for early and late spring (Fig. 4).

## DISCUSSION

This study demonstrates that remote sensing can be used to examine the role of land use change in leading to shifts in phenology, a result often attributed to climate change (*Walther et al., 2002*; *Parmesan, 2006*; *Tylianakis et al., 2008*; *Ovaskainen et al., 2013*).

Major differences in land surface phenology and bee community spatio-temporal distributions exist between urban, agricultural, and natural land use types. The phenology of land surface vegetation in human-altered landscapes is largely out of synchrony with surrounding California grassland natural areas. In addition to different patterns of phenological change, different land use types exhibit variability in the ranges and standard deviations of their land surface vegetation phenology. Further, these patterns correlate with a spatio-temporal bee distribution data.

In California grasslands, green vegetation is largely driven by temperature and rainfall, resulting in a large burst of blooms in the spring, and by the end of the summer, there are few floral resources available (*Chiariello, 1989*). However, in human-altered landscapes, landscaping and water irrigation patterns are likely even stronger influences. Vegetation in urban areas is highly diverse, and selected for aesthetic and convenience reasons. Urban areas, while having less green space, often have many exotic ornamental plants which are supplemented with water and nutrient inputs that allow for an extended flowering season. As a result, urban areas are characterized by low, but constant, floral resources throughout the year. Agricultural areas have large patches of dense, often homogeneous, floral resources that will fluctuate greatly from early spring to the end of the summer. In this system, stone fruit orchards are in flower in the spring, but throughout the summer there are other crops in flower such as alfalfa, tomato, corn, and bell peppers (*Guise, 2011*).

This asynchrony in land surface phenology between neighboring land use types is similar to what *Buyantuyev & Wu (2012)* found in the desert landscapes of the southwestern United States. They found the timing of highest peaks of vegetation indices to be different between land use types, which they attributed to a decoupling in the urban sites from the local climatic drivers (*Buyantuyev & Wu, 2012*). It is likely that a similar scenario occurs in California, with grasslands juxtaposed against urban and agricultural areas that have different vegetation types and additional inputs.

Beyond the different timing in vegetation indices, it is important to note the significantly different standard deviation and ranges of pixels of the same land use type. Natural land use types were quite similar to one another given their relatively small standard deviations and ranges over time. However, in the human-altered landscapes, particularly agricultural, there was much higher variation. In other words, even though natural areas can be considered patchy, they are not nearly as patchy as urban and agricultural areas on the same scale. One reason for the enhanced patchiness is that in urban and agricultural landscapes, there are many different land owners and management decisions being made on a relatively micro-scale, resulting in a wide diversity of vegetation types being selectively planted and cared for in different ways across the landscape. Instead of vegetation type shifting on the scale of a few kilometers, it might actually differ on the scale of a few meters. Such extreme patchiness of vegetation can have many implications for organisms dependent on floral resources. Additionally, varying patterns in vegetation in urban landscapes have been found to be closely tied to socio-economic factors (*Luck, Smallbone & O'Brien, 2009*), suggesting an important factor to consider when exploring biodiversity in human-altered landscapes. We were able to detect varying patterns in vegetation even

though the resolution of the remote sensing data was limited to 250 m. As remote sensing data technology advances, we will be able to explore more fully these patterns.

There are clearly different patterns of bee abundance between land use types over the course of the year. Differences in land surface phenology between land use types were detected through remote sensing, although the relationship between remote sensing vegetation indices and the bee community is complex due to the role of seasonality and major differences in vegetation between land use types. The strongest relationship between vegetation indices and bee abundances were in natural sites, but sites classified as natural also exhibited high collinearity between seasonality and vegetation indices, making it difficult to tease apart the differences. The overlapping points between seasons in the urban and agricultural sites (Fig. 4) mean that vegetation indices are less tied to seasonality.

Despite this lack of collinearity with seasonality in human-altered landscapes, there were still significant patterns between vegetation indices and bee abundance, indicating the potential to use remote sensing to detect certain aspects of biodiversity. Further exploration into the relative contributions of different types of vegetation within each land use type and between time of year would contribute to the strength of applying these findings more broadly. In addition, building on the idea of "scaling up from urban gardens" (*Goddard, Dougill & Benton, 2010*), it would be valuable to see if patterns change when the larger surrounding area of each location is investigated. No significant relationship was detected between vegetation indices and species richness as a general indicator, although focusing on certain species or functional groups may highlight which are more vulnerable to changes in phenological patterns (*Moussus et al., 2011*).

In this dataset, bee collections were made approximately every two months, yet certain species peaked in abundance at different times between land use types. This lack of synchrony in peak abundance between land use types could be the result of two possibilities: either bees are moving (further than expected) between land use types in search of better resources, or localized population structuring is occurring between different land use types based on differences in timing of emergences. Perhaps on a finer temporal scale of collections, for example, on the same frequency as MODIS composite products (every 16 days), these subtleties in timing between land use types could be better captured. Additionally, the role of time lags should be further explored by investigating floral resource availability. Models with increasing time lags fit better than those without time lags, although this may still be tied to the strong effect of seasonality in natural areas. However, several other studies have shown the importance of time lags between changes and ecological responses (*Caperon, 1969*; *Metzger et al., 2009*). In this case, it is likely the result of challenges in using vegetation indices as proxies for floral availability. Sometimes extreme floral abundance can lead to an underestimation of vegetation biomass (*Shen et al., 2010*). Perhaps time delay is the result of plants experiencing green up through leaf growth before producing floral reproductive structures. A better understanding of species associations with vegetation indices would further improve predictive power of utilizing remote sensing data to predict species distributions.

These findings have implications for restoration of human-altered landscapes. A great amount of interest and resources go into creating "green spaces" and restoring patches of land within urban and agricultural matrices (*Linehan, Gross & Finn, 1995*; *Rudd, Vala & Schaefer, 2002*; *Snep et al., 2006*). However, most restoration practices are based on studies in more natural land use contexts (*Dobson, Bradshaw & Baker, 1997*; *Purcell, Friedrich & Resh, 2002*; *Crossman et al., 2007*). Due to the much higher degree of patchiness in human-altered landscapes and changes in phenological patterns, restoration goals and strategies may need to be altered when working in human-altered landscape contexts. Additionally, this study highlights the importance of temporal differences in human-altered landscapes.

Understanding how human-altered landscapes impact species distributions and interactions is critical as land use change accelerates globally. In order to overcome previous limitations when using remote sensing to estimate biodiversity, it is first important to further understand the dynamics of vegetation type, phenology, and the ecology of interacting taxa. These results clearly indicate that the phenology of vegetation in different land use types are not synchronized, and vegetation indices created through remote sensing can predict bee community abundance. Such findings suggest the potential to use remote sensing to estimate the communities of other taxa beyond bees, as well as provide a new way of understanding the ecological challenges of urbanization and agriculture associated with phenological differences.

### Funding

Funding was provided by the Margaret C. Walker Fund, Calaveras Big Trees Association, UC Berkeley Graduate Division Grant, Harvey I. Magy Memorial Scholarship, Berkeley Natural History Museums GK-12 Fellowship, and Berkeley Connect. The funders had no role in study design, data collection and analysis, decision to publish, or preparation of the manuscript.

### Grant Disclosures

The following grant information was disclosed by the authors:
Margaret C. Walker Fund.
Calaveras Big Trees Association.
UC Berkeley Graduate Division Grant.
Harvey I. Magy Memorial Scholarship.
Berkeley Natural History Museums GK-12 Fellowship.
Berkeley Connect.

### Competing Interests

The author declares there are no competing interests.

## Author Contributions

- Misha Leong conceived and designed the experiments, performed the experiments, analyzed the data, contributed reagents/materials/analysis tools, wrote the paper, prepared figures and/or tables, reviewed drafts of the paper, processed, prepared, and identified specimens.
- George K. Roderick contributed reagents/materials/analysis tools, reviewed drafts of the paper.

## Supplemental Information

Supplemental information for this article can be found online at http://dx.doi.org/10.7717/peerj.1141#supplemental-information.

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
