# Peer review of "Remote sensing captures varying temporal patterns of vegetation between human-altered and natural landscapes"

_PeerJ, doi:10.7717/peerj.1141_

## Round 0.1 · original submission · Minor Revisions

· Academic Editor

Minor Revisions

This is an interesting manuscript and topical. Both reviewers are positive but I also have a number of other comments. Generally you needed to be clearer about where you are getting some information, and direct the reader to evidence/tables. Also I feel there is some important information missing.

Major comments
Very little information about sites. The number of sites is only mentioned in the bee collection section. Assume 8 sites per landtype..., Maybe a table with site information would be useful, but at least re-working the methods section a little.

There were no results on the bee community! Even basic results are not reported: number of species/genera and their abundance, how this is broken down across landtypes or seasons. Given the 'core' part of your study is linking remote sensing with bees, this is needed. You have mentioned l269 about specific peaks in abundance but no data/evidence provided. The bee community is also likely to vary in time.

Table 1 and 2 - why is there no data for natural sites?

Table 2 not included in text. Also the headings in table are not clear replace as those in table 1

Figure 4 is very important, linking bee abundance with remote sensing but its hard to read, and more needs to be made of the slopes or regression lines?, with statistical values. It may be difficult to have these on the figure, but in another table

Minor comments
throughout - delete double spaces
l49 - bracket needed - events).
l63 - where was this study?
l67 - replace text - . "Therefore, there..."
lines 179-187 use of natural, agricultural, urban but without "sites" afterwards, and then later in results, use of "sites" and also "areas", be consistent
l199 "appeared", be more assertive, based on results
l201 use of the word significantly-please direct reader to the evidence for this, table?
fig1 - labels within boxes are very hard to read yet they are very important to show the progression
fig2 - figures are not labelled as a, b, c
fig4 - bold first sentence. label and values on axes too small. remove legend from figure and place it in figure legend.

·

Basic reporting

In general, this study is well-designed and the manuscript well-written. The Abstract & Introduction provide a clear introduction to the field & justification for this study.

Line 53: should ‘product’ be ‘produced’?

Line 80: ‘Also, biodiversity may be too broad...’ is a bit vague. Do you mean to say ‘focusing on biodiversity may be too broad’, or perhaps ‘using ‘biodiversity’ as a study organism may be too broad’

Line 86-7: ‘..and nuts that require bee pollination.’ should read ‘...and nuts, requiring bee pollination.’ Also, perhaps you could change ‘bee’ to ‘insect’, as many other wild insect pollinators are critical to food production besides bees.

Line 91: ‘with external inputs that results in’ should read ‘with external inputs, resulting in’

Line 211-214: The sentence ‘I found major differences...’ and the sentence ‘The phenology of land surface vegetation...’ seem to contradict each other. Also, there are a few typos in the second sentence – ‘in’ should appear before ‘human-altered’ and ‘are’ should be deleted

Line 224: ‘homogenous’ should be ‘homogeneous’

Line 256 (and others): ‘colinearity’ should be ‘collinearity’

Line 300: ‘to estimate other taxa’. What about other taxa needs to be estimated? Do you mean perhaps ‘to estimate the distributions of other taxa’ or the ‘abundance’?

Figure 4: please clarify that you are talking about ‘bee abundance’ in the caption

Experimental design

Lines 148-155: For readers who are new to remote sensing techniques, could you please provide a brief explanation of what EVI and NDVI are and why they are appropriate for your study questions? This is important for readers to be able to interpret the results. I think EVI is more suited to assessing seasonal patterns?? So how/why does EVI differ from NDVI and why did you use these? Any information that will help readers understand the relationships shown in your results will help.

Lines 172-174: Please also provide some detail (either here or in the results) about whether the collecting method affected results? There is a huge difference in the number of hours that traps were open for across years (4 vs 24 hrs), and this would have significantly affected the relative abundances you collected. How does the model you used genuinely account for this discrepancy; or would it be better to omit the 2010 data from the analysis?

Validity of the findings

Line 239: I think the difference in patchiness arises at different scales, e.g. natural ecosystems are usually more patchy at finer scales, agricultural landscapes usually more patchy at larger scales, while patchiness in urban landscapes may vary, depending on whether you are talking about an industrial, suburban residential or inner-city business district. The relevance of this to your study should be clarified here.

Line 248: Do you mean that you detected patterns of correlation with socio-economic variability in your analyses? This wasn’t clear from your results.

Lines 281-283: What about orchard trees (e.g. some of the crops in your study landscape), many of which mass flower before they produce leaves? Would this over/under estimate high or low EVI/NDVI?

Line 286-293: I think this is a really important point and needs a bit more explanation. Restoration of ecosystems within managed landscapes should incorporate peripheral effects arising from differences in management and phenology compared to unmanaged ecosystems. How do your results inform this approach?

Additional comments

This is an interesting study that takes a novel approach to understanding how landscape-scale shifts in phenology influence pollinator abundance. Using remote sensing data to explore relationships between biodiversity and vegetation is a relatively new method and has rarely been used to study pollinator communities. In particular, understanding how pollinator communities are influenced at the landscape-scale by managed land uses (e.g. mass-flowering crops or urban areas) is an important ecological question – using remote sensing to explore the abrupt spatial and temporal changes in floral resource availability that arise between natural and agricultural or urban land uses is an interesting way to approach this question. I have some queries about Methods and a few minor suggestions for edits related to clarity of expression and the need for more detail in some parts (see detailed comments).

Reviewer 2 ·

Basic reporting

This manuscript investigates how vegetation phenology varies in a human-altered California grassland landscape, and whether or not these changes in phenology correlate with those of the bee community that depends on floral resources.
The writing is clear enough, however the manuscript contains a number of spelling and grammatical errors which should be carefully revised before acceptance. Furthermore, Figures could benefit from a more detailed legend to make them more accessible. The time stamps in Figure 1 are hard to read, as are the lines in Figure 2, which makes them difficult to interpret. Figure 4 appears to include trendlines for the different land types, however there seems to be no explanation in the Figure legend or any values which describe the strength of the effect, it may be helpful to put these directly in the Figure.
The raw data of the NDVI and EVI data appears to be available online (Reference 1), and the bee community data will be uploaded to the Dryad data depository (but does not appear to be publicly available yet).

Experimental design

Overall the question is interesting and the approach of using remote sensing techniques, including multi seasonal variation in phenology, could help estimating biodiversity for other taxa beyond bees. The question is clearly defined and meaningful.
The study appears to be appropriately designed and executed, showing major differences in land surface phenology and bee community distributions between urban, agricultural, and natural land use types. Methods are sufficiently described and research has been conducted in conformity with prevailing ethical standards.

Validity of the findings

The dataset appears to be sufficiently large enough to answer the research question, however since each of the sites were sampled multiple times, I wonder though if a repeated measures approach could be more appropriate for the analysis as data points of the same sampling location may not be independent.
A potential shortcoming is the limited resolution of the remote sensing technology, which has been apropriately acknowledged in the manuscript. It would be great to see GPS coordinates of the actual sampling sites.

---

## Round 0.2 · accepted · Accept

· Academic Editor

Accept

I am writing to inform you that your manuscript, "Remote sensing captures varying temporal patterns of vegetation between human-altered and natural landscapes" has been accepted for publication.